# Human Exposure to Naturally Occurring *Bacillus anthracis* in the Kars Region of Eastern Türkiye

**DOI:** 10.3390/microorganisms12010167

**Published:** 2024-01-14

**Authors:** Fatih Buyuk, Hugh Dyson, Thomas R. Laws, Ozgur Celebi, Mehmet Doganay, Mitat Sahin, Les Baillie

**Affiliations:** 1Department of Microbiology, Faculty of Veterinary Medicine, Kafkas University, Kars 36300, Türkiye; fatihbyk08@hotmail.com (F.B.); mitats@hotmail.com (M.S.); 2CBR Division, Defence Science and Technology Laboratory, Porton Down, Salisbury SP4 0JQ, UK; ehdyson@dstl.gov.uk (H.D.); trlaws@dstl.gov.uk (T.R.L.); 3Department of Medical Microbiology, Faculty of Medicine, Kafkas University, Kars 36100, Türkiye; ozgurcelebi36@hotmail.com; 4Department of Infectious Diseases, Faculty of Medicine, Lokman Hekim University, Ankara 06530, Türkiye; mehmet.doganay@lokmanhekim.edu.tr; 5Faculty of Veterinary Medicine, Kyrgyz-Turkısh Manas Unıversıty, Chingiz Aitmatov Campus, Djal, Bishkek 720038, Kyrgyzstan; 6School of Pharmacy and Pharmaceutical Sciences, Cardiff University, Cardiff CF10 3NB, UK

**Keywords:** *Bacillus anthracis*, occupational exposure, clinical infection, human serology, Kars region, endemic anthrax, protective antigen, lethal factor

## Abstract

Environmental contamination with *Bacillus anthracis* spores poses uncertain threats to human health. We undertook a study to determine whether inhabitants of the anthrax-endemic region of Kars in eastern Türkiye could develop immune responses to anthrax toxins without recognised clinical infection. We measured anti-PA and anti-LF IgG antibody concentrations by ELISA in serum from 279 volunteers, 105 of whom had previously diagnosed anthrax infection (100 cutaneous, 5 gastrointestinal). Of the 174 without history of infection, 72 had prior contact with anthrax-contaminated material. Individuals were classified according to demographic parameters, daily working environment, and residence type. All villages in this study had recorded previous animal or human anthrax cases. Stepwise regression analyses showed that prior clinical infection correlated strongly with concentrations at the upper end of the ranges observed for both antibodies. For anti-PA, being a butcher and duration of continuous exposure risk correlated with high concentrations, while being a veterinarian or shepherd, time since infection, and town residence correlated with low concentrations. For anti-LF, village residence correlated with high concentrations, while infection limited to fingers or thumbs correlated with low concentrations. Linear discriminant analysis identified antibody concentration profiles associated with known prior infection. Profiles least typical of prior infection were observed in urban dwellers with known previous infection and in veterinarians without history of infection. Four individuals without history of infection (two butchers, two rural dwellers) had profiles suggesting unrecognised prior infection. Healthy humans therefore appear able to tolerate low-level exposure to environmental *B. anthracis* spores without ill effect, but it remains to be determined whether this exposure is protective. These findings have implications for authorities tasked with reducing the risk posed to human health by spore-contaminated materials and environments.

## 1. Introduction

*Bacillus anthracis*, the etiological agent of anthrax, is a bacterium that primarily infects herbivores and humans only occasionally as accidental hosts. The bacterium spends most of its life cycle as a spore that is highly resistant to harsh environmental conditions [1]. Uptake by a susceptible host triggers spore germination to the vegetative form with the expression of a range of virulence factors, which include an anti-phagocytic capsule and a tripartite protein toxin. The toxin proteins are produced in significant amounts soon after germination is initiated [2]. The tripartite toxin complex [3,4,5] consists of two enzymatically active subunits [5] called lethal factor (LF) and edema factor (EF), which are transported across host cell membranes by a pore-forming subunit called protective antigen (PA). This protein is so named due to its ability to stimulate a protective antibody-based immune response and is the principal protective immunogen in a range of animal and human vaccines [6,7,8,9,10,11,12,13,14,15].

On the death of an infected host, the bacterium reverts to its spore form, which can persist in the environment and remain infectious to further susceptible hosts for many years [1,16,17]. Little is known about the manner in which animals become infected or indeed the factors that determine intra-herd susceptibility. The currently held view is that animals grazing on anthrax-contaminated land ingest spores along with soil or spiky vegetation, creating entry points for bacteria via lesions in the gastrointestinal tract. In arid regions, it is possible that animals which graze or take dust baths near anthrax carcass sites could become infected by inhaling airborne dust laden with anthrax spores. Animal to animal transmission appears to be a rare event and when it does occur, it is believed to be mediated by biting flies [18]. Humans normally contract anthrax directly or indirectly from animals or animal products [19,20,21,22,23,24,25]. Naturally occurring anthrax presents in one of three clinical forms: cutaneous, ingestional (oropharyngeal and gastrointestinal), and inhalational [26]. The most common form of the disease, cutaneous anthrax, occurs when spores enter the body through a break in the skin. Infection commonly occurs at sites in contact with animal products, such as the neck and shoulders due to carrying infected carcasses or forearms, wrists, and hands following flaying hides or butchering meat. Gastrointestinal infection occurs in parts of the world where the nutritional value of meat outweighs the perceived risks of serious illness from its consumption [27,28,29].

Anthrax occurs worldwide, with active foci reported in Australia, China, Kazakhstan, India, Pakistan, North America, sub-Saharan Africa, Europe, and the Anatolian peninsula [30]. There has been a steady decline in the incidence of animal and human cases of anthrax in Türkiye from 6.8 per million in 1990 to 4.1 per million in 2005 [22]. The majority of these cases occurred around the cities of Kars and Erzurum, which are located in the northeast of the country [20]. Both cities are centres for animal trade and have access to large international commercial roads, thereby potentially facilitating the spread of anthrax spores. In 2008, an animal vaccination programme, supported by education of livestock owners, was initiated by Kafkas University. These endeavours, together with the introduction of a government compensation scheme [31] in 2012, are believed to be responsible for the decrease in animal and human cases reported in the Kars region.

Humans can contract anthrax following inhalation of airborne spores of *B. anthracis*. Historically, this occurred due to occupational exposure to spore-contaminated wool and goat hair in textile mills, where inhalation anthrax was known as Woolsorter’s Disease. However, not all exposed individuals developed clinical disease [32,33]. Furthermore, occupational exposure [34] to aerosolised spores can stimulate anti-PA antibody responses in workers with no history or clinical signs of infection [32,35,36]. These studies demonstrated that healthy individuals who work in contaminated industrial environments can develop antibodies against anthrax toxin antigens, which further supports the hypothesis that *B. anthracis* can cause subclinical infection [36].

This project aimed to determine whether individuals living in the Kars region of eastern Türkiye, where anthrax is endemic, could develop an immune response to anthrax toxin antigens without any history of clinical symptoms. Two main groups of volunteers were recruited for this study: individuals known to have previously had clinical anthrax infection and other individuals with no history of infection who had varying degrees of exposure risk as assessed by their predominant daytime environment.

## 2. Methods

### 2.1. Study Design

This study was carried out in the anthrax-endemic region of Kars in eastern Türkiye, according to a protocol approved by the Ethics Committee of the Faculty of Medicine, Erciyes University, Kayseri. The primary objective was to ascertain whether individuals exposed to *Bacillus anthracis* spores during daily life could develop serological responses to bacterial toxin antigens without evidence of active infection. All individuals taking part in this study were aged at least 18 and were long-term residents of Kars province.

Patients previously diagnosed with anthrax and treated in Kafkas University Hospital were identified from clinical records and contacted individually. The number found and agreeing to take part was estimated to comprise around 50% of the former anthrax patient population in the Kars region.

In addition to patients with a previous episode of diagnosed anthrax infection, volunteers were sought from each of the following groups:(1)Individuals living in a village in Kars province that had a known history of animal or human anthrax infection and who were regularly involved in animal husbandry (rural dweller group);(2)Butchers who worked in an anthrax-endemic area;(3)Shepherds who worked in close contact with animals;(4)Leather workers who worked with potentially contaminated material;(5)Veterinarians (including animal health technicians) working in an anthrax-endemic area;(6)Individuals living in Kars city or one of five towns in the region: Arpaçay, Digor, Selim, Susuz, or Sarıkamış (urban dweller group);(7)Laboratory staff working in the Pathology and Microbiology departments at Kafkas University in Kars.

Systematic identification of representative samples of potential participants within these groups was not possible. Individuals were therefore approached opportunistically as determined by their accessibility; if interested, they were informed about the study. We estimate that in the butcher, laboratory staff, leather worker, and veterinarian groups, those enrolled comprised around 80% of the individuals in those occupations living in the areas surrounding Kars city. In contrast, the number of shepherds and rural and urban dwellers whom it was possible to approach comprised very small proportions of their total respective populations in the Kars region.

After a briefing about the nature and purpose of the study, fully informed consent was obtained from each volunteer. A 15 mL sample of whole blood was then drawn from each individual by venepuncture; serum was separated by centrifugation and aliquots were stored at −80 °C prior to analysis. Demographic details were recorded for each participant, together with any prior history of infection or exposure to material potentially contaminated with anthrax spores. 

### 2.2. Demographic and Clinical Information

Age, sex, and occupation were recorded for each of the 279 study participants. Occupational categories, defined to reflect each individual’s predominant daytime environment, were butcher [n = 48), laboratory staff [n = 12], leather worker [n = 5], rural dweller [n = 108], shepherd [n = 29], urban dweller [n = 31], and veterinarian (including animal technicians) [n = 46]. Also noted was the time spent in these occupational categories, which for rural dwellers and urban dwellers was considered to be the same as their age. No study participant had been vaccinated against anthrax. Descriptive statistics are shown in Appendix A for two groups of participants: (1) with known previous anthrax infection, (2) with no history of anthrax infection.

The information recorded about each individual’s residence comprised the category (city, town, or village), altitude (ascertained from Google Earth), population size, and administrative district within Kars province. Each administrative district consists of a concentrated population centre—either Kars city or one of the five towns (Arpaçay, Digor, Sarıkamış, Selim, Susuz) together with surrounding villages. Population figures (all dated 2012) for Kars city and the five towns were taken from Wikipedia. Figures for the villages are estimates derived from www.citypopulation.de/en/Türkiye/kars (accessed on 16 February 2022) for 2012 using the 2/3 point between the values for 31 December 2009 and 31 December 2013, as blood samples were taken between 14 April 2012 and 15 December 2012. The Citypopulation database includes only those villages with populations of >750; all other villages were therefore assumed to have populations of ≤750.

Kars region has a humid continental climate with significant seasonal and diurnal temperature variation. Winters are very cold (mean temperatures around −9 °C; sometimes −30 °C), with four months of snow cover on average; springtime thaws often produce substantial local flooding, while summers are warm (mean temperatures around 18 °C; maximum 35 °C) and often humid [37]. Due to the potential importance of different climatic conditions in environmental anthrax cycles, the website www.mindat.org (accessed on 16 February 2022) was explored to ascertain Köppen climate types for each residence location. However, all areas for which information was available were listed as Dfb; in addition, locations for which no classification was recorded had surrounding or adjacent sites classified as Dfb, so they were assumed to have this climate type also. (In the Köppen climate type classification, D indicates a continental climate with ≥1 month averaging <0 °C and ≥1 month averaging >10 °C; f indicates no dry season; b indicates a warm summer.)

For individuals with previously confirmed anthrax infection, the anatomical sites affected were classified as follows: Face, including eye, cheek, and jaw [n = 6, 5.7%]; digits (i.e., fingers and/or thumb) [n = 33, 31.4%]; hand [n = 21, 20.0%]; wrist [n = 15, 14.3%]; arm [n = 18, 17.1%]; unspecified cutaneous [n = 7, 6.7%]; gastrointestinal (GI) tract [n = 5, 4.8%]. The probable route of infection was not recorded for the majority of patients, but was assumed to be direct or indirect contact with, or consumption of, contaminated animal products.

### 2.3. Serological Analyses

#### 2.3.1. Optimised Enzyme Linked Immunosorbent Assay (ELISA) Method

Antigen-specific IgG responses in each serum sample were determined using the following protocol:(1)The first 10 columns of each 96-well ELISA plate (Biosigma S.r.l, Coma, Italy) were coated with 100 µL recombinant antigen at 2 µg/mL in PBS. Both recombinant protective antigen (Lot 17115A5B) and lethal factor (Lot 1722B11B) were purchased from Quadratech Diagnostics Ltd. (Eastbourne BN21 3AW, UK);(2)A standard curve was constructed by adding human IgG (Sigma Code: I4506; Sigma Aldrich, Dorset SP8 4XT, UK) to the last two columns (11 and 12) to achieve a range of concentrations (0.5, 0.25, 0.125, and 0.0625 µg/mL), with each concentration represented in triplicate;(3)Following coating, plates were washed three times with 300 µL of PBS + 0.1% Tween-20 (PBST) per well (Wellwash Microplate Washer, Cat. No. N15777, ThermoFisher, Vantaa FI-01621, Finland) and then incubated at 37 °C for 1 h to block the wells;(4)Test samples (columns 1–8), positive control serum (column 9), and negative control serum (column 10) were defrosted and serially diluted 1 in 2 down the plate with PBST, with the first well containing a 1:10 dilution;(5)Following incubation, plates were washed three times with PBST;(6)A total of 100 µL anti-human IgG mouse monoclonal antibody (Jackson ImmunoResearch Lab. Inc., Cambridgeshire CB7 4EX, UK. Code: 209-035-088, Lot: 100924), diluted 1:1000 in PBST, was added to each well. Plates were then incubated for 1 h at 37 °C;(7)Following incubation, plates were washed three times with PBST using an automated plate washer;(8)Substrate solution was prepared as follows: 0.7 g sodium phosphate dibasic (Sigma Code: S5136-100G) and 0.5 g citric acid (Sigma Code: 27102) were dissolved in 100 mL of deionised water. One ABTS substrate tablet (Sigma Code: 1001235282, A9941, Lot: 071M8224V) was then added to this solution. Finally, 2.5 µL H_2_O_2_ (Sigma Code: H1009-100ML) was added to 10 mL of the substrate solution prior to use;(9)A total of 100 µL ABTS reagent with H_2_O_2_ was added to each well and the plates incubated at 37 °C for 30 min. The reaction was stopped by adding 100 µL 2% SDS (Sigma Code: L4390-100g) in deionised water, and the plates were read at 405 nm (SpectraMax Plus384 Absorbance Microplate Reader, Molecular Devices).

#### 2.3.2. Determination of IgG Concentration 

Serum samples were tested three times against each specific antigen. Samples were run on different occasions so that variance was randomly distributed between replicates. An Excel macro was written to batch process the results. Briefly, the macro selects the first dilution of each sample that gives an OD value less than 0.6 (approximately in the centre of the linear response of the assay system) and subtracts the OD value of the negative serum sample of the same dilution. It then calculates the antibody concentration by reference to a linear regression performed on the mean values of the triplicate human IgG standards. These values were then collated to give a mean antibody concentration for each sample with each antigen; all values are expressed as µg/mL.

### 2.4. Statistical Analyses

Microsoft Excel was used to collate source data, some of which were manipulated in Excel prior to statistical analysis. SPSS software V27.1 was employed to generate descriptive demographic statistics and perform the following analyses: stepwise linear regression, multiple regression, correlation (Spearman’s and Pearson’s methods), and linear discriminant factor analysis (a supervised learning technique). Graphs were prepared using Graphpad PRISM V8.0.

#### 2.4.1. Missing Data Points

There were several missing points in the collated dataset; such missing data points can prevent effective statistical modelling. When points were missing because the relevant information had not been obtained, the strategy adopted was to interpolate the population mean. The advantage of this approach is that differences are minimised, thus decreasing the potential for false positives. When missing points arose due to non-applicable categories (for example “time since infection” for individuals with no known infection), an arbitrary number greater than the maximum or lower than the minimum (depending on the bias of the variable) was assigned. The substitutions employed are listed in Appendix A. For villages with population recorded as <750, the value 750 was used.

#### 2.4.2. Data Transformations

Categorical variables were transformed into dummy variables. Examination of continuous variables using quantile–quantile plots indicated that time since infection and population exhibited lognormal properties, so these were transformed prior to analysis. Both sets of ELISA data were shown to be exponentially distributed and were therefore transformed to the natural logarithm (log_e_) for analysis. Some ELISA results were negative (due to a control-based adjustment), so the transformation included the addition of a correction factor slightly greater than the minimum value for each set of results, according to the formula x = log_e_(x + k), where k = (minimum + 0.001). Similar values were given by each set of ELISA replicates, although one clear outlier of a triplicate of analyses for anti-LF IgG was excluded. The mean of the transformed values for the two or three ELISA results for each antibody was taken for each participant. 

#### 2.4.3. Stepwise Linear Regressions

Antibody concentration datasets were analysed by separate stepwise linear regressions. Progression of the steps in each model was predicated on -F-tests comparing models where variable inclusion was set at *p* < 0.05. The results of the regression for anti-PA IgG were arrived at through eight steps, and for anti-LF IgG through five steps. The validity of the model was assessed by the examination of residuals and comparing these to a Gaussian distribution. Correlations to consider as covariates were determined using Spearman’s method. Pearson’s method was used to assess whether there might be a correlation between antibody concentrations. Categorical variables were considered using stepwise binary logistic regression involving likelihood ratio tests, where variable inclusion was set at *p* < 0.05. 

#### 2.4.4. Linear Discriminant Analysis

The data were explored in an attempt to identify a profile of antibody concentrations that best indicated a true previous anthrax infection. Linear discriminant analysis was employed to explore how these two measurements might best be used in combination to discriminate between infected and non-infected groups. A composite discriminant factor optimised for identifying previous infection that used corresponding values for both antibody concentrations was then derived. This discriminant factor could predict previous infection with good sensitivity and specificity, as was demonstrated by its receiver operator characteristic (ROC) curve, which had an AUC of 0.898 (Section 3.2).

#### 2.4.5. Stepwise Multiple Regressions Using Discriminant Factor

This discriminant factor was then used to consider which recorded variable might indicate when an infection had occurred previously but had not been recognised. A stepwise multiple regression analysis was performed using only data from individuals with no history of anthrax infection together with the discriminant factor. This process was inverted to identify the characteristics of individuals with known previous infection who had antibody concentration profiles least typical of past infection. A stepwise multiple regression analysis was performed using only data from individuals with known previous anthrax infection together with the discriminant factor.

## 3. Results

### 3.1. Factors Associated with Anti-PA and Anti-LF IgG Concentration

Factors found to correlate independently with anti-PA IgG concentrations are shown in Appendix A, and those correlating with anti-LF IgG concentrations in Appendix A. Confirmed previous anthrax infection was associated with high concentrations of IgG antibodies against both PA (Figure 1A, *p* < 0.001) and LF (Figure 2A, *p* < 0.001). Specific occupational categories were found to be associated with anti-PA IgG concentrations (Figure 1B): positively in the case of butchers (*p* = 0.004), but negatively for veterinarians (*p* = 0.001) and shepherds (*p* = 0.007). 

Time since infection (Figure 1D) was negatively associated (*p* = 0.013), and duration of continuous risk of exposure (Figure 1E) positively associated (*p* = 0.024) with anti-PA IgG concentration. Town residence was negatively associated (*p* = 0.012) with anti-PA IgG concentration, while village residence was positively associated (*p* < 0.001) with anti-LF IgG concentrations. Individuals from Arpaçay district (Figure 1C) had relatively elevated (*p* = 0.024) anti-PA IgG concentrations, while those from Sarıkamış district (Figure 2C) had relatively elevated (*p* < 0.001) anti-LF IgG concentrations. 

Anti-LF IgG concentrations were lower (*p* = 0.042) in individuals who had had anthrax infections that involved only digits (i.e., fingers and/or thumbs), compared to those with infection at other anatomical loci (Figure 2E).

There was a statistically significant (*p* = 0.029) association with a very small effect size (coefficient 1.25 × 10^−8^) between sample date and anti-LF IgG concentration (Figure 2D). There were no obvious correlations (where r < −0.5 or r > 0.5) between these factors and other explanatory variables apart from intuitively obvious associations of:
(1)the categories of living in a village with being a rural dweller by occupation;(2)time since infection with confirmed previous infection. 

### 3.2. Derivation of Optimised Composite Discriminant Factor

Correlation between the concentrations of the two antibodies in the whole dataset was relatively weak (Figure 3A) with r = 0.6 (0.52–0.67 95% CI), suggesting that there might be added benefit to using both parameters to identify a pattern characteristic of prior anthrax infection. Linear discriminant analysis was therefore employed to explore how these two measurements might best be used in combination to discriminate between infected and non-infected groups. The output indicated that values for anti-PA IgG concentration contributed proportionately 1.6 times more than the corresponding anti-LF IgG concentration in identifying previous infection. 

A composite discriminant factor optimised for identifying previous infection, that used corresponding values for both antibody concentrations, was then derived. This discriminant factor could predict previous infection with good sensitivity and specificity as demonstrated by its Receiver Operator Characteristic (ROC) curve, which had an AUC of 0.898 (Figure 3B). This provided only a slight increase in predictive accuracy compared to anti-PA IgG concentration alone, which had an AUC of 0.885 (Figure 3C). In contrast, the minor contribution made by anti-LF IgG concentration to the composite discriminant factor is demonstrated by its ROC curve, which had an AUC of only 0.615 (Figure 3D).

### 3.3. Antibody Concentration Profiles Typical of Previous Infection

The results of the stepwise multiple regression analysis using data from individuals with no history of anthrax infection together with the composite discriminant factor are shown in Appendix A. They indicate that antibody profiles typical of previous infection are negatively associated (*p* < 0.001) with being a veterinarian (Figure 4A) and positively associated (*p* = 0.002) with the duration of continuous exposure risk (Figure 4B). Thus, veterinarians were least likely to have had an unrecognised infection, but the longer an individual is continuously at risk of exposure to anthrax spores, the more likely they are to have developed an unrecognised infection.

The results of the stepwise multiple regression analysis using only data from individuals with known previous anthrax infection together with the discriminant factor are shown in Appendix A. They indicate that, amongst the previously infected group, antibody concentration profiles least typical of past infection were significantly associated with the occupational group of urban dwellers (*p* = 0.007, Figure 4C), residence in Kars Central district (*p* = 0.035, Figure 4D), and increasing time since infection (*p* = 0.038, Figure 4E). 

### 3.4. Identification of Individuals with Probable Unrecognised Previous Infection

Corresponding values for anti-PA IgG and anti-LF IgG concentrations are shown in Figure 5 for individuals with no history of infection and also for those with known previous infection. The probability of belonging to the known previous infection group, as determined by linear discriminant analysis, is indicated by the size of circle for each individual. 

The probability values resulting from the linear discriminant analysis were examined for the group with no history of infection to determine whether, based on these antibody responses, any individuals might have previously had an unrecognised infection. Four individuals within this group had antibody concentrations indicating a >90% probability of previous anthrax infection (Figure 5A and Figure 6A). Two were butchers and two were rural dwellers. Anti-PA and anti-LF IgG concentrations for these four individuals are shown in Figure 6 with those of the other individuals in their occupation groups, both with and without history of prior anthrax infection, confirming that their antibody responses were typical of subjects with known previous infection. Demographic characteristics and serum antibody concentrations for these four individuals are listed in Appendix A.

## 4. Discussion

The numbers of reported cases of animal and human anthrax in Türkiye have continued to decline in recent years due to the efforts of the Turkish government. During the period 2007 to 2019, there were 8320 reported animal infections. The majority (32.5%) occurred in cattle, with approximately 20% of these cases being reported in the Erzurum/Kars region [38]. This reduction in animal cases is probably responsible for the reduction in reported human cases over the same time period. 

Anti-toxin antibodies are detectable in human serum for prolonged periods after exposure to anthrax toxin antigens during infection [39,40,41], and also following vaccination [42,43,44]. As might be expected, we found detectable anti-PA and anti-LF IgG antibodies in serum from all 105 participants with prior confirmed anthrax infections, which had occurred up to 25 years before the start of our study. Of these individuals, 100 presented with a single episode of cutaneous infection, while the remaining 5 had contracted the gastrointestinal form of the disease.

The sites of infection reported in our study can be explained by the handling and processing of contaminated animal products. Digits (31%), hands (20%), wrists (14%), and arms (17%) were the commonest sites of cutaneous infection, while the face (6%) was affected infrequently. Gastrointestinal tract infection (5%) was assumed to have arisen from the consumption of contaminated meat. A similar pattern was found in a study of 58 patients with cutaneous anthrax admitted to hospital in Ankara [21], the predominantly affected sites being hands (39%) and fingers (29%), followed by forearms (12%), eyelids (7%), and neck (3%). Exposure to contaminated animal products was the most likely source of infection in our study, but it is possible that some individuals were exposed to *B. anthracis* spores through contact with contaminated soil, although clinically confirmed cases are extremely rare [45].

Based on the published literature, cases of re-infection of previously infected individuals are rare [46] and were not seen in our study. One possible explanation for this could be the stimulation of a protective immune response as it is known that anthrax-infected individuals mount cellular- and antibody-based immune responses to a range of antigens [39,40,41]. These responses include the production of toxin-neutralising antibodies, which in animal studies have been correlated with protection, but they wain with time [41] and it is unclear whether they are sufficient to prevent re-infection during an individual’s lifetime. Toxin-specific antibody responses have also been observed in asymptomatic individuals working in environments contaminated with anthrax spores, suggesting that continued exposure to spores can also induce an immune response in the absence of clinical infection [41]. It remains to be determined whether the antibodies generated by such exposure are protective.

When we compared the antibody results of our study participants, we observed considerable overlap in the distributions of anti-PA and anti-LF IgG responses in individuals with confirmed previous infections and those individuals who had no history of infection. Values at the lower end of each distribution, particularly near the lower level of quantification (LLQ) of the ELISAs, could be attributed to cross-reactivity [47] with other environmental antigens [41], while those at the top of the distribution could be due to unrecognised previous infection. A possible explanation for values in the middle of the antibody concentration distributions is unrecognised exposure to levels of anthrax spores too low to cause clinical infection; similar conclusions were drawn from studies in a New Hampshire goat-hair-processing mill [32] and in a Belgian factory that processes imported wool and goat hair [36].

Potential sources of *B. anthracis* spores in our study were likely to be both environmental and occupational. Individuals at risk of environmental exposure are those who live and work in villages (rural dwellers), where animals that have died of anthrax were butchered on open ground or buried next to residential buildings without prior decontamination. Spore exposure could occur sporadically or even daily, depending on the individual’s routine activities and their place of residence in relation to the contaminated site. Individuals at risk of occupational exposure in an anthrax-endemic region are those who work with animals (shepherds, veterinarians) or animal products (butchers, leather workers); they would be expected to encounter sick animals or contaminated material on occasion without necessarily being aware of the fact. In contrast, those who live and work in towns or cities in occupations unrelated to animals (urban dwellers) or work with infectious material in microbiological containment laboratories (laboratory staff) are much less likely to be at risk of exposure to anthrax spores.

The results of the stepwise regressions support these expectations to some extent, with the occupation of butcher being predictive of elevated anti-PA IgG, village residence being linked to elevated anti-LF IgG, and town residence being predictive of low anti-PA IgG concentrations. Reassuringly, the category of laboratory staff was not a predictor for high concentrations of antibodies against either antigen. Somewhat surprising was the finding that being a shepherd or a veterinarian were strong predictors for low anti-PA IgG concentrations. It could be that these groups of individuals have greater awareness of the disease and so adopt measures to reduce the potential for infection when handling sick animals. Butchers, on the other hand, may be unaware that some of the slaughtered animals they handle have been infected prior to death, and as a result do not adopt measures to minimise their infection risk. Indeed, an association between butchering and handling infected animals and a positive immune response to *B. anthracis* toxins has previously been reported [19,21]. Furthermore, although the sample sizes were relatively small, the findings of a recent study in Khyber Pakhtunkhwa province, Pakistan [48], were consistent with our results in suggesting that in anthrax-endemic areas, butchers are at relatively high risk and veterinarians at much lower risk of unrecognised exposure to anthrax spores. These observations all underline the importance of ongoing education schemes, such as those undertaken by Kafkas University veterinarians, to raise awareness of the risks involved in handling contaminated animal products and to improve burial procedures for infected animal carcasses.

Both anti-PA and anti-LF IgG have been shown to contribute independently to the neutralisation of anthrax lethal toxin activity [43,44], suggesting that these two anti-toxin antibodies may have synergistic actions. In our study, there were differences between these two antibodies in that village residence was a predictor for elevated anti-LF IgG but not for anti-PA IgG concentration, while none of the parameters associated with elevated anti-PA IgG were predictors for elevated anti-LF IgG concentrations. In addition, linear discriminant analysis indicated that the optimal discriminant factor for past anthrax infection was a composite of both anti-PA and anti-LF concentrations, although the major contribution was made by anti-PA concentrations. The findings of our study therefore suggest that antibodies to PA and LF may be generated under different conditions; this is consistent with observations that patterns of cellular and humoral immunity against LF and PA differ in various contexts [39,40].

Overall, the results of the linear discriminant analysis were complementary to the stepwise regression results. Thus, in individuals with no history of previous infection, veterinarians were significantly less likely than the other occupational groups to have antibody concentration profiles typical of infection, while the longer the period of continuous exposure risk an individual had, the more likely they were to exhibit these profiles. In individuals with known previous anthrax infections, urban dwellers, residents of Kars Central district (which includes Kars city and its surrounding villages), and those with long time intervals between infection and sampling were all likely to have antibody profiles least typical of past infection. Antibody profiles therefore appear to differ between occupational groups and change with time since infection and the duration of continuous risk of exposure.

Based on the combined PA and LF results, discriminant analysis identified four individuals with no history of clinical infection who had antibody concentrations suggestive of past exposure to anthrax. Two of these individuals were butchers, a group shown in our study to have elevated anti-PA IgG, and two were rural dwellers, a group with elevated anti-LF IgG concentrations in our study. While cross-reaction to one toxin component is possible, cross-reaction to both in the same individual in the absence of exposure to lethal toxin seems unlikely. Based on these results, therefore, we would argue that an antibody response to both toxin components is suggestive of prior exposure to the pathogen.

In conclusion, healthy humans appear to be able to tolerate low-level exposure to environmental *B. anthracis* spores without ill effect, but it remains to be determined whether such exposure can generate any protective immunity. Confirmation by further studies that low-level spore exposure generates a degree of protective immunity in humans would have significant implications for authorities responsible for making safe environments that have been contaminated with *B. anthracis* spores. The challenge such contamination poses is well illustrated by the aftermath of the 2001 mail attacks in the United States, where clean-up work guided by a strategy of no detectable spores proved to be difficult and extremely costly [49].

## 5. Study Limitations

This was a real-world observational study for which participants were recruited according to their availability rather than by selection from the population of Kars province on the basis of pre-determined demographic characteristics. Although it was therefore not possible to calculate the population statistics of the various parameters studied, the statistical models used were able to explore the data gained to make comparisons within the context of the study population. This enabled us to draw valid conclusions that can provide the basis for further studies.

The following specific limitations were noted:(1)There were missing data points for some parameters for a few individuals in this study, but the statistical approaches used to account for these gaps in a dataset are well recognised as being able to satisfactorily eliminate any uncertainty caused;(2)Infected individuals were treated with antibiotics and as a consequence the resulting immune response may not fully reflect what happens when an infection resolves without any external intervention;(3)While an immune response indicates past exposure to a pathogen, it cannot be assumed that the individual would be protected from future infection. Further studies are required to determine whether this is indeed the case in the context of low-level exposure to anthrax spores. These studies should include, inter alia, measurement of lethal toxin neutralisation using the TNA assay [50,51], which is recognised by regulatory authorities as a surrogate for anthrax vaccine efficacy.

## Figures and Tables

**Figure 1 microorganisms-12-00167-f001:**
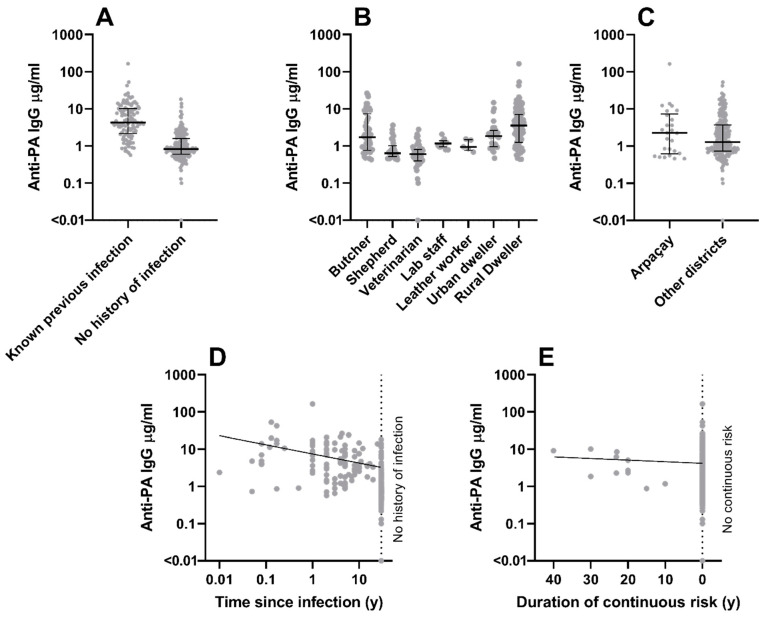
**Factors indicated as likely correlates with anti-PA IgG concentration in a multiple regression.** Potential roles are shown for confirmed previous anthrax infection (panel **A**), different occupational groups (panel **B**), residence in Arpaçay district (panel **C**), time interval between infection and sampling (panel **D**), and duration of continuous exposure risk (panel **E**). Each data point represents one individual. For categorical variables, median +/− interquartile ranges are given. For continuous variables, a regression line indicates the direction of effect.

**Figure 2 microorganisms-12-00167-f002:**
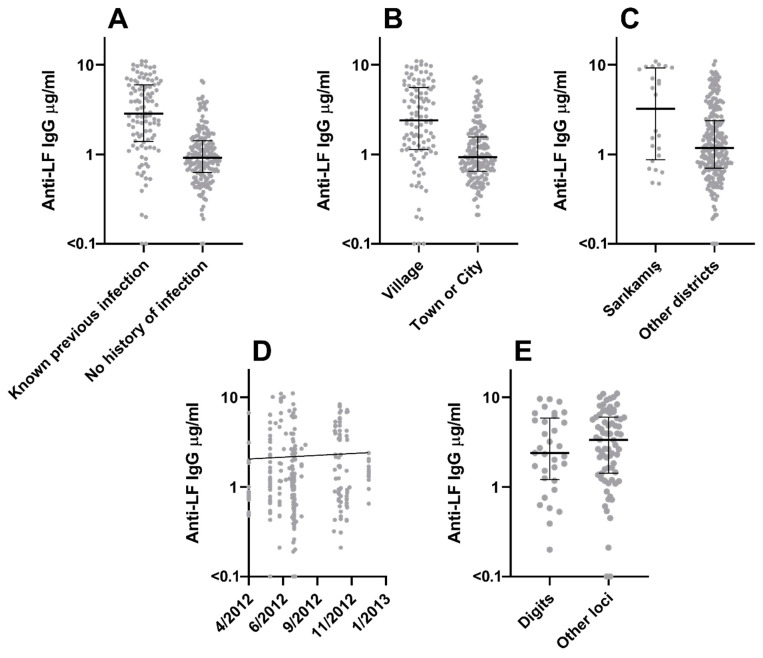
**Factors indicated as likely correlates with anti-LF IgG concentration in a multiple regression.** Potential roles are shown for confirmed previous anthrax infection (panel **A**), residence category (panel **B**), residence in Sarıkamış district (panel **C**), sample date (panel **D**), and infection only on fingers or thumbs (panel **E**). Each data point represents one individual. For categorical variables, median +/− interquartile ranges are given. For continuous variables, a regression line indicates the direction of effect.

**Figure 3 microorganisms-12-00167-f003:**
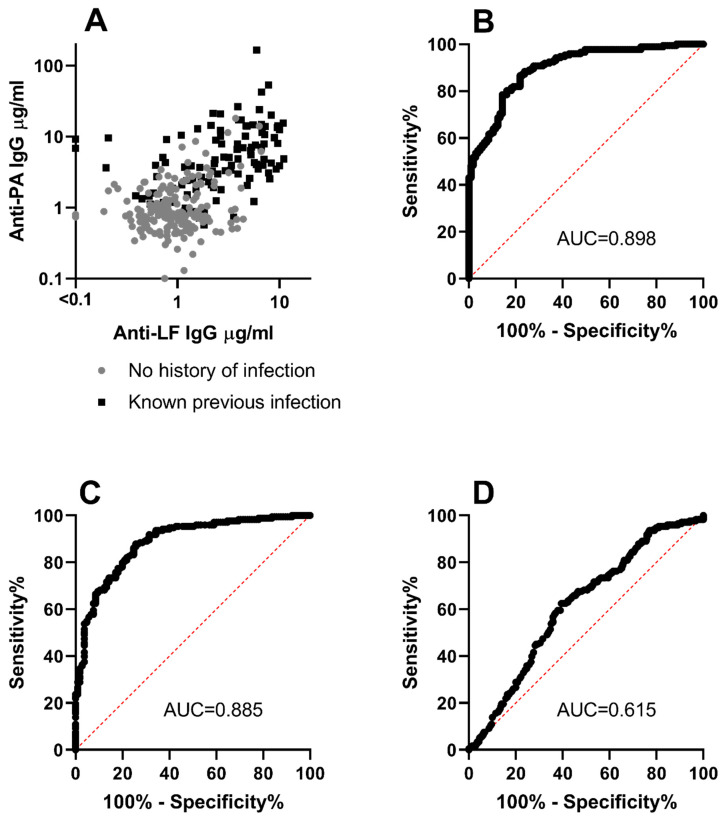
**Results of analysis to identify antibody concentration profiles typical of previous infection.** (Panel **A**): each data point represents one individual; concentrations of anti-PA and anti-LF IgG did not correlate well [r = 0.6 (0.52–0.67 95% CI)]. Discriminant scores were estimated and exhibited favourable receiver operator characteristics (Panel **B**) in identifying samples from individuals with previous confirmed anthrax infection. ROC curves for anti-PA IgG alone and anti-LF IgG alone are shown in (Panels **C** and **D**).

**Figure 4 microorganisms-12-00167-f004:**
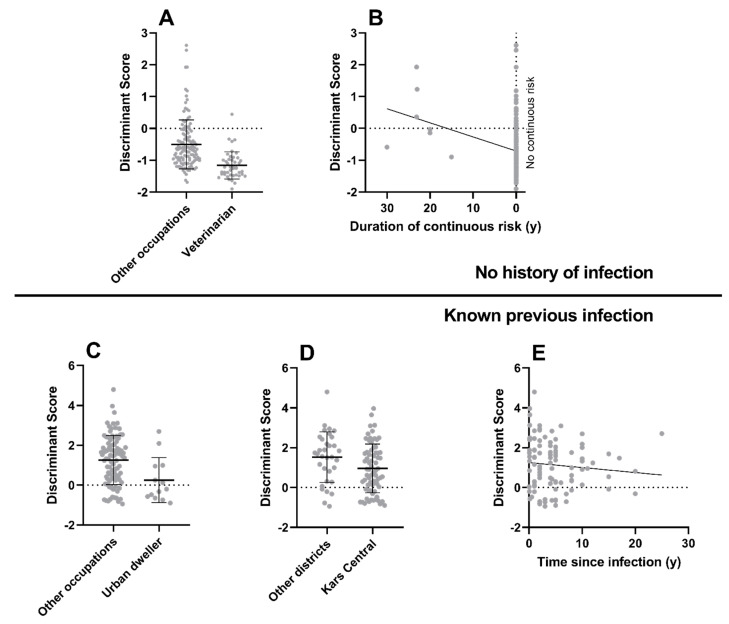
**Results of analysis to identify factors associated with presence and absence of previous anthrax infection.** A discriminant factor, which exhibited favourable receiver operator characteristics in identifying results from individuals with known previous infection, was used in two multiple regression analyses to identify: (**i**) Factors that might indicate when an infection had occurred previously but had not been recognised (this analysis used only data from individuals with no history of anthrax infection); (**ii**) Characteristics of individuals with known previous infection who had antibody concentration profiles least typical of past infection (this analysis used only data from individuals with known previous anthrax infection). The first regression showed the discriminant factor to be negatively correlated with the occupation of veterinarian (Panel **A**) and positively correlated with the duration of continuous exposure risk (Panel **B**). The second regression showed the discriminant factor to be negatively correlated the urban dweller occupation group (Panel **C**), residence in Kars Central administrative district (Panel **D**), and increasing time since infection (Panel **E**), indicating that individuals with these characteristics all had antibody concentration profiles that were least typical of past infection. For categorical variables, median +/− interquartile ranges are given. For continuous variables, a regression line indicates the direction of effect. Each data point represents one individual.

**Figure 5 microorganisms-12-00167-f005:**
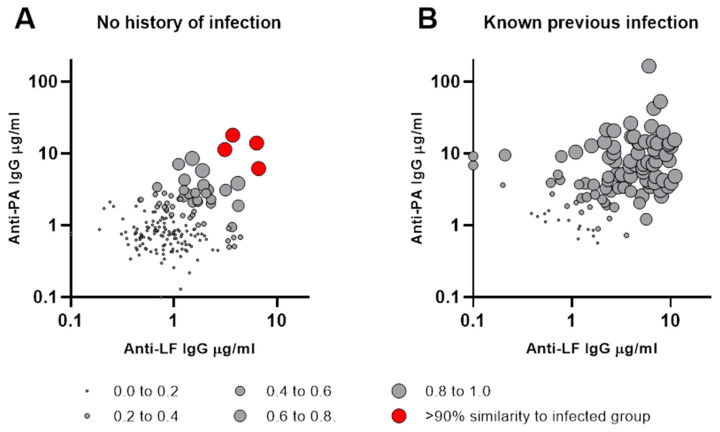
**Probability of membership of past infection group as assessed by linear discriminant analysis.** Corresponding anti-PA IgG (y-axis) and anti-LF IgG (x-axis) concentrations are shown for individuals with no history of infection in (Panel **A**) and those with known previous infection in (Panel **B**). The probability of belonging to the known previous infection group, as determined by linear discriminant analysis, is indicated for each individual by the size of circles in (Panels **A** and **B**).

**Figure 6 microorganisms-12-00167-f006:**
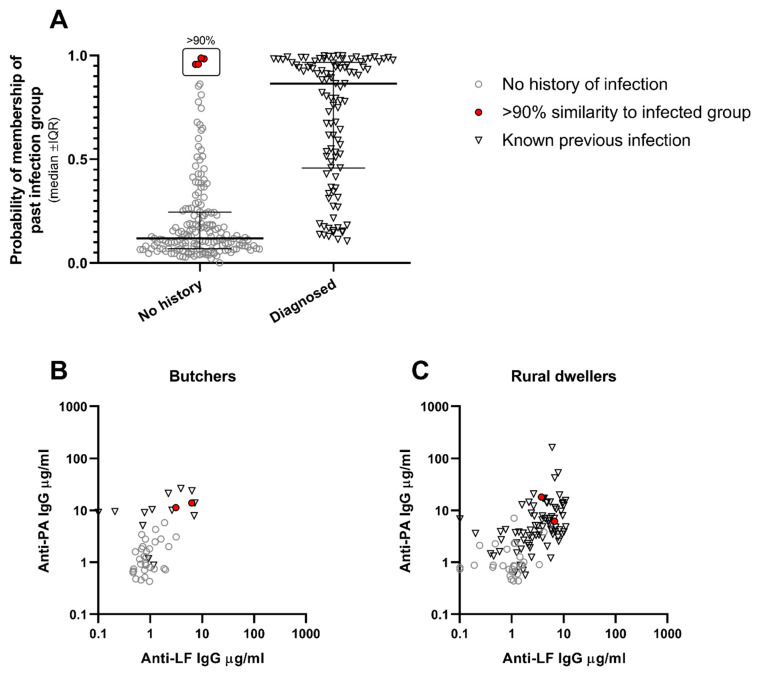
**Evidence of undiagnosed previous cases of anthrax infection within the dataset.** (Panel **A**) shows the likelihood of belonging to the group with known previous infection, as calculated by linear discriminate analysis. This discriminate analysis used concentrations of anti-PA and anti-LF IgG antibodies to ascertain optimal values to determine whether an individual had previously had an anthrax infection. Four individuals with no recorded history of anthrax had antibody concentrations indicating a >90% probability of previous infection. Two were butchers and two were rural dwellers (Appendix A). Their results for both anti-PA and anti-LF IgG concentrations are shown together with all other individuals in their occupations, both with and without history of prior anthrax infection ((Panel **B**) for butchers, (Panel **C**) for rural dwellers).

## Data Availability

Data are contained within the article and Appendix A.

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
