# Peer review of "Human Exposure to Naturally Occurring Bacillus anthracis in the Kars Region of Eastern Türkiye"

_microorganisms, 2024, doi:10.3390/microorganisms12010167_

Round 1

Reviewer 1 Report

Comments and Suggestions for Authors

The manuscript Human exposure to naturally occurring Bacillus anthracis in the Kars region of eastern Türkiye, written by Buyuk F et al, brings to the attention an interesting aspect, of humans having antibodies against anthrax, without prior infection.

The manuscript is well written with detailed protocols, however I have the following recommendations:

-references are not in the right format of the journal

- at the end of the manuscript the Conflict of interest is not written and the funding is put in acknowledgements 

- references are too old

- introduction is too long, please shorten the introduction and add info about the immune response to anthrax and also about the classification of the forms of anthrax

- in the discussion you could introduce information about similar articles with identified antibodies in people not infected

Comments on the Quality of English Language

only small changes required

Author Response

Thank you for your kind comments and for pointing out how the paper can be improved. We have made the following changes and additions as you suggested:

1) Change the reference citations from Author (year of publication) to Number [X] according to the style used by Microorganisms.

2) Conflict of Interest statement has been included and Funding information moved from Acknowledgements to Funding section.

3) Nine more recent references have been added, increasing the total to 40.

4) We have shortened and simplified the Introduction, and included information about the clinical presentations of anthrax and humoral immune responses to its toxin antigens. 

5) We have added information about previous reports of anti-toxin antibodies being found in individuals with no history of clinical infection, that suggest that sub-clinical infection can occur. These range from the early studies in New Hampshire textile mills in the 1950s, reported by Norman and Plotkin in 1960, to studies in a Belgian wool factory in the 200s, reported by Wattiau and Kissling in 2008-12. However, rather than include these in the Discussion, we felt they were more appropriately placed in the Introduction. We hope that this is satisfactory.

Reviewer 2 Report

Comments and Suggestions for Authors

The article by Fatih Buyuk1, et al., which discusses Human Exposure to Naturally Occurring Bacillus anthracis, is interesting, original, and suitable for publication in this journal. The article is well-written and organized; and discussing important issue. Investigation of anthrax in blood of previously infected and contact persons in important to take an overview about the situation in the area of study. The figures and tables are informative and provide the results in a good wan. However, some minor comments must be addressed before publication. The introduction can be improved by giving an overview about the situation in Europe and Asia, as Turkey is Eurasian country. You may use the following articles for knowledge DOI:10.3390/microorganisms11051294 and Doi: https://doi.org/10.51585/gjm.2023.1.0021 and maybe include more studies. I think citation is not as known mdpi style, please check with the editorial office. In discussion, the official registered date on anthrax in Turkey at WHO or WOAH or FAO better to be included in the discussion to give an overview about the official situation in the country.  conclusion and limitataions of the study are acceptable:

Author Response

Thank you for your kind comments and helpful suggestions.

We have added more information to the Introduction about the prevalence and changing anthrax infection rates in different parts of the world, and have added a paragraph at the beginning of the Discussion about the recent incidence of human and animal anthrax cases in Türkiye. Following comments by yourself and the other Reviewer, we have actually added another nine new references (to make a total of 40) overall, which we believe has significantly improved the paper.

You correctly identified that the citation style we used (Author & year of publication) was not the one used by Microorganisms (Number [X]), so we have made the necessary changes. Thank you for pointing this out.